**PLOS** **Pathogens**

# Outdoor roaming of owned cats elevates risk of zoonotic pathogen exposure: A global synthesis

**Amy G. Wilson**[1]*, **Scott Wilson**[1,2], **Peter P. Marra**[3], **David R. Lapen**[4]

**1** Department of Forest and Conservation Sciences, University of British Columbia, Vancouver, British Columbia, Canada, **2** Wildlife Research Division, Environment and Climate Change Canada, Edmonton, Alberta, Canada, **3** Earth Commons Institute, Department of Biology, Georgetown University, Washington, District of Columbia, United States of America, **4** Agriculture and Agri-Food Canada, Science and Technology Branch, Ottawa, Ontario, Canada

* amy.wilson@ubc.ca

## Abstract

Domestic animals play a central role in pathogen transmission at the human–wildlife interface. Domestic cats, in particular, are uniquely consequential in disease spillover dynamics due to their global distribution, large, human-subsidized free-roaming populations, and high contact rate with humans, domestic animals, and wildlife. However, the extent to which human ownership and management mitigate this spillover risk remains a key knowledge gap. To address this gap, we conducted a global systematic review and quantitative synthesis of the prevalence and diversity of zoonotic pathogens in indoor-only, outdoor-owned (roaming unsupervised), and unowned (feral or stray) cats. Our dataset comprised 174,064 individuals from 88 countries, representing 124 pathogen species, 97 of which are zoonotic. Using generalized linear models within a Bayesian framework and rarefaction analyses, we show that ownership provides limited protection against zoonoses when owned cats have unsupervised outdoor access. Outdoor-owned cats were 3–5 times more likely to carry zoonotic pathogens than indoor-only cats, and, notably, had infection odds statistically equivalent to those of feral cats, despite receiving presumed veterinary care and feeding. Feral cats carried the highest pathogen diversity, however, outdoor-owned cats still harbour 1.5 times the helminth richness of indoor cats, highlighting their potential as effective bridges for pathogen spillover. With approximately 62% of owned cats roaming freely worldwide, and rates exceeding 90% in some regions, these findings reveal a major yet overlooked route of zoonotic risk. Public health and One Health frameworks have traditionally focused on feral cats; however, our results highlight the need to explicitly incorporate owned outdoor cats into zoonotic disease prevention strategies by restricting unsupervised roaming and promoting responsible ownership practices. Without such integration, current frameworks risk overlooking a pervasive and preventable pathway for pathogen transmission at the human–wildlife–domestic animal interface.

**Data availability statement:** Data compiled for this study is provided in the supplementary information.

**Funding:** Funding for this project was provided by Agriculture and Agri-Food Canada's Environmental Change One Health Observatory (ECO2) project (J-002305) to DRL. The funders had no role in study design, data collection and analysis, decision to publish, or preparation of the manuscript. Salary support was received from Agriculture and Agri-Food Canada to DRL and AGW.

**Competing interests:** The authors have declared that no competing interests exist.

## Author summary

Cats are among the most common companion animals worldwide and are often allowed to roam freely outdoors, bringing them into close contact with people, other domestic animals, and wildlife. While these interactions raise well-known conservation concerns, they also expose cats to a wide range of pathogens. Disease risk is commonly associated with feral cats, but the relative risk for owned cats allowed to roam outdoors unsupervised has remained unclear. To address this question, we compiled data from studies conducted globally, encompassing a broad range of pathogens relevant to human health. We found that cats with outdoor access were more likely to carry infections than indoor-only ats, and that their risk of infection was comparable to that of feral cats. This similarity suggests that outdoor access leading to hunting or contact with wildlife and other free-roaming animals can rapidly negate the protective effects of ownership. Because owned cats also have frequent close contact with people, infection acquired outdoors may increase opportunities for onward exposure to owners and the broader public, including through environmental contamination. Together, these results show that how people manage their pets plays a major role in shaping disease transmission between wildlife, domestic animals, and humans, highlighting that strategies designed to reduce the ecological impacts of free-roaming cats can simultaneously deliver substantial benefits for both public health and biodiversity.

## Introduction

In the One Health framework, which emphasizes the interconnectedness of human, animal, and environmental health, wildlife is frequently identified as a primary source of emerging zoonotic diseases [1]. The emergence and spillover of these pathogens are intensified by anthropogenic disturbances such as habitat loss, land use change, climate disruption, and toxicant release, all of which alter epidemiological dynamics and increase opportunities for transmission across the wildlife–human interface [2,3]. A frequently overlooked but consequential outcome of these disturbances is the expansion and redistribution of domestic animal populations into wildlife habitats.

Within altered landscapes, free-roaming domestic animals occupy a unique ecological position, functioning as high-contact bridge hosts that connect wildlife, human, and environmental pathogen pools, thereby facilitating pathogen circulation and amplification across otherwise discrete host communities [4,5]. By moving repeatedly between human-dominated spaces and wildlife habitats, domestic species can integrate multi-host transmission networks and reshape zoonotic risk at the human–wildlife boundary. Human subsidization further supports unnaturally high densities of domestic animals at the wildlife-human interface, positioning them as transport hosts, amplifiers, and reservoirs for zoonotic pathogens of wildlife origin [6,7]. Domestic animals may also be underrecognized as sources of zoonoses, as

pathogen-sharing between domestic animals and humans exceeds that between humans and wildlife for a broad range of pathogens, including viruses [4], helminths [8], and ectoparasites [9,10]. This close epidemiological connection is reinforced by evidence that the ability to infect companion animals is the strongest predictor of zoonotic potential in helminth pathogens [11].

Among domestic animals, dogs and cats have high societal value and emotional familiarity, resulting in frequent close contact with humans. Emotional familiarity with companion animals can suppress risk perception, allowing zoonotic exposure pathways to persist largely unmitigated [12,13]. Consequently, free-roaming dogs and cats function as high-contact bridge hosts, linking wildlife reservoirs with human populations and amplifying opportunities for cross-species pathogen transmission. Free-roaming cats play a particularly significant role at the human–wildlife interface due to their frequent interactions with wildlife. Globally, over 2,000 wildlife species have been documented as prey for domestic cats [14], and in the United States and China alone, cat-associated wildlife mortality is estimated at billions annually [15,16]. Many of these prey species have limited or no natural contact with humans, making each predation event a potential pathway for rare or emerging wildlife pathogens to be transmitted to humans or other domestic animals. For example, a single-owned free-roaming cat returned prey that were infected with two previously undescribed viruses with zoonotic potential [17,18].

Domestic cats exist along a spectrum of human management, ranging from contained indoor cats to completely free-roaming feral cats. Understanding the main determinants of zoonotic disease risk is essential for risk mitigation, as different management categories of domestic cats are unlikely to be equivalent in their zoonotic risk. Indoor cats, which do not roam unsupervised, are isolated from the majority of zoonotic pathogens. Feral cats spend their entire lives outdoors, and through hunting and other close interactions with wildlife, experience broad exposure to all pathogens. The intermediate group consists of outdoor-owned cats that are allowed to roam at rates that vary spatially and temporally [19]. Although these outdoor-owned cats receive consistent feeding, varying levels of veterinary care, and partial confinement, they still interact with wildlife, making it unclear whether this partial management meaningfully reduces the risk that outdoor-owned cats act as conduits for zoonoses of wildlife origin.

Here, we investigate how variation in cat free-roaming influences both the prevalence and diversity of zoonotic pathogens, as a test of how the management of domestic animals alters pathogen amplification pathways at the human–wildlife interface. We hypothesized that outdoor access is the predominant driver of exposure to zoonotic pathogens, but that ownership would moderate this exposure risk by reducing the duration of outdoor access relative to feral cats. Specifically, we predicted that free-roaming owned cats (i.e., owned cats allowed outdoor access) would harbour fewer and less diverse pathogens than free-roaming feral cats (e.g., stray or feral cats), but more than indoor-only cats, reflecting continued pathogen exposure through hunting and roaming. Because ownership-associated correlates, such as anthelmintic treatment, are rarely reported as continuous variables across studies and do not necessarily covary with free-roaming behaviour, we treated cat lifestyles as discrete categories that represent broad positions along a free-roaming continuum. To test our hypothesis, we conducted a systematic review and meta-analysis of studies reporting the prevalence of zoonotic pathogens in domestic cats. By extracting lifestyle information from these studies, we classified cats along a roaming gradient from absent (indoor-only) to moderate (owned but allowed to roam freely) to high (feral). This dataset enabled us to robustly test the association between management and zoonotic disease risk, considering both pathogen prevalence and diversity.

## Results

Our literature search yielded 2,422 studies, of which 604 met the inclusion criteria (S1 Fig), representing 174,067 individual cats from 88 countries. Of these studies, 435 provided management information, yielding a sample size of 86,301 individuals. Across 86 country-level estimates, the average proportion of owned cats with outdoor access was 62% (range: 3% – 92%; Fig 1). One hundred and twenty-four pathogens were identified at the species level

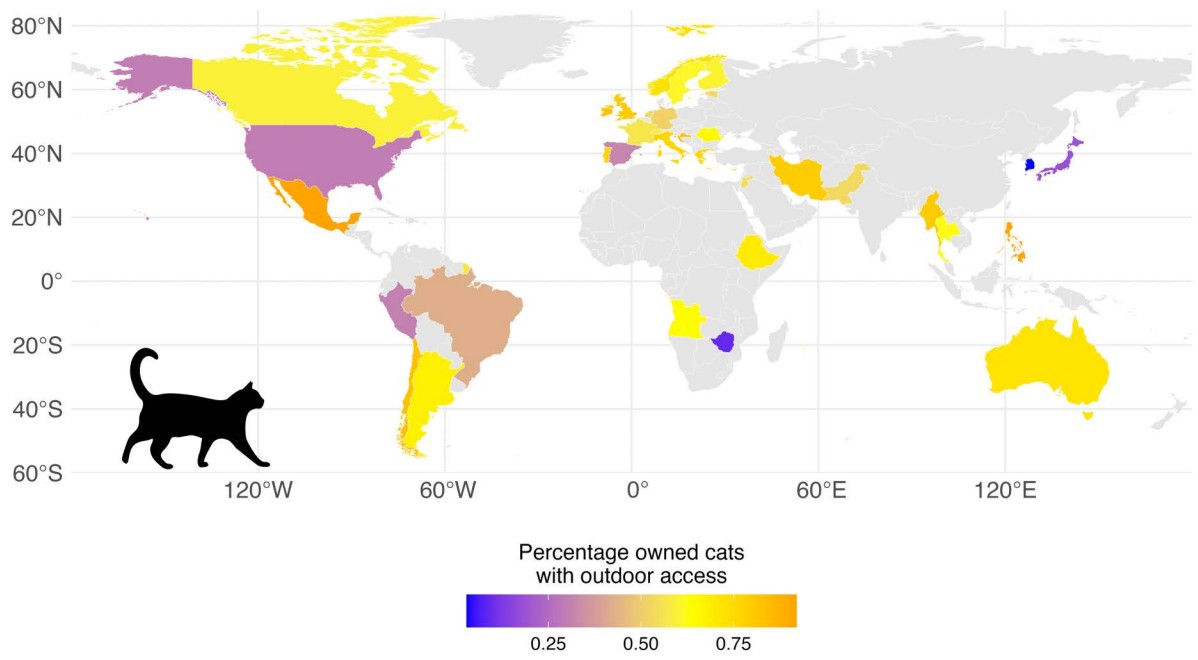

**Fig 1. The proportion of owned cats reported to have a free-roaming lifestyle was summarized across 86 estimates from 38 countries in a global dataset.** Cat silhouette obtained from PhyloPic and licensed under CC0 1.0 Universal Public Domain Dedication. The basemap was made with Natural Earth (https://www.naturalearthdata.com).

and 29 at the genus level, of which 97 and 22 have been identified as zoonotic (S1 Table). We did not find support for our hypothesis that moderate management through ownership lowers the risk of pathogen infection for outdoor-owned cats compared to feral cats. The posterior mean log-odds of infection pooled across all pathogens were lowest for owned indoor cats ($\beta_{indoor}$ = -3.21, 95% Credible Interval (CrI): -3.73 – -2.68). While outdoor-owned cats exhibited significantly higher pooled infection risk than indoor cats as predicted ($\beta_{outdoor-owned}$ = 1.12, 95% CrI: 0.81 – 1.42), we did not find any difference in the infection risk for outdoor-owned cats compared to feral cats ($\beta_{feral}$ = 1.24, 95% CrI: 0.91 – 1.55), contrary to predictions. Pooled prevalence estimates showed a similar pattern, with outdoor-owned (18.24%, 95% CI: 17.90 – 18.59) and feral cats (18.04%, 95% CI: 17.82 – 18.26) exhibiting substantially higher overall positivity than indoor cats (8.26%, 95% CI: 7.95 – 8.59). When expressed as odds ratios (OR), the odds of infection with any pathogen were three times higher in outdoor-owned cats than in indoor cats (OR = 3.08, 95% CI: 2.19 – 4.04), while outdoor-owned cats had comparable odds relative to feral cats (OR = 1.14, 95% CI: 0.85 – 1.46) (Fig 2). The effect estimates remained consistent when analyses were restricted to studies published after 2000 and 2010 (S5 Table).

For individual pathogens with sufficient data for pathogen-specific analysis, outdoor-owned cats had significantly higher odds for infection than indoor cats for *Toxoplasma gondii* (OR = 3.2, 95% CI: 2.12 – 4.42) and *T. cati* (OR = 4.89, 95% CI: 2.84 – 7.22), but this effect was more variable for Bartonella (OR = 3.52, 95% CI: 1.13 – 7.30) and Leptospira (OR = 3.46, 95% CI: 0.65 – 7.67). Outdoor-owned and feral cats had odds ratios overlapping 1 for *T. gondii* (OR = 1.3, 95% CI: 0.89 – 1.78), *T. cati* (OR = 1.34, 95% CI: 0.66 – 2.16), Bartonella (OR = 1.58, 95% CI: 0.52 – 3.02) and Leptospira (OR = 0.88, 95% CI: 0.36 – 1.50). For Giardia, comparable odds of infection were found between indoor and outdoor-owned cats, and between outdoor-owned and feral cats, at 1.37 (95% CI: 0.84 – 2.00) and 1.15 (95% CI: 0.44 – 1.91), respectively. For

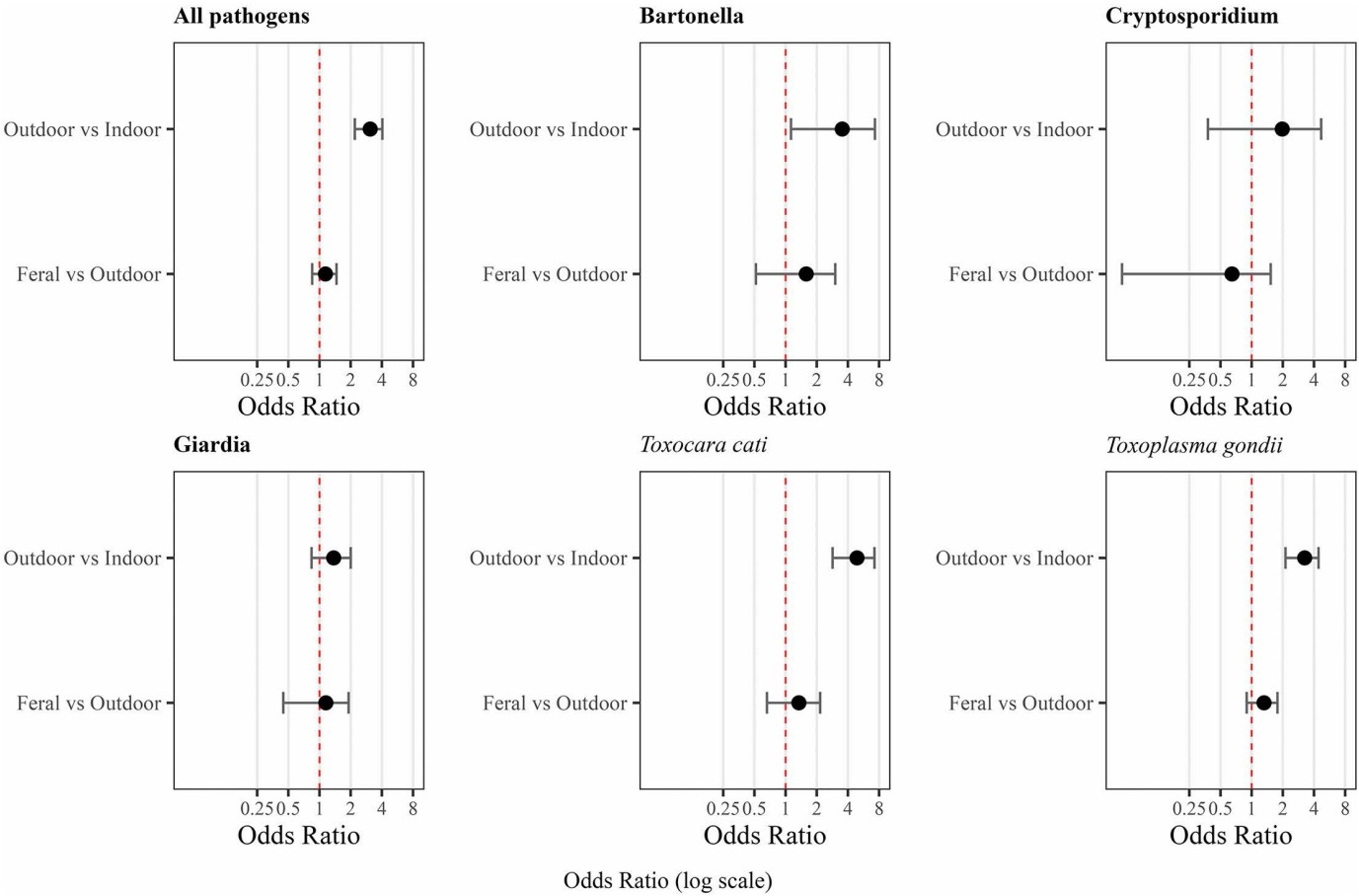

**Fig 2. Odds ratios with 95% credible intervals for pathogen exposure, compiled across the global data set for all pathogens and separately for Bartonella, Cryptosporidium, Giardia, *Toxocara cati*, *Toxoplasma gondii* and Leptospira, for outdoor-owned cats relative to feral cats and for outdoor-owned cats relative to indoor cats.**

Cryptosporidium, high uncertainty was present in the odds ratios between indoor and outdoor-owned cats (OR: 1.97, 95% CI: 0.37 – 4.69) and between outdoor-owned and feral cats (OR: 0.64, 95% CI: 0.05 – 1.53). Even higher uncertainty was present for Ancylostoma between outdoor cats and indoor cats (OR: 2.23, 95% CI: 0.09 – 6.31) and outdoor-owned cats and feral cats (OR: 7.23, 95% CI: 0.6 – 17.71).

The diversity and richness of helminth pathogens varied across feral, indoor, and outdoor-owned cat assemblages, and we found some support for the hypothesis that cat ownership reduces the diversity of pathogens to which domestic cats are exposed. The most common helminth pathogens were the zoonotic *T. cati* and Ancylostoma, which together accounted for 35.6% and 14.6% of all infections, respectively. Observed species richness was highest in feral cats (38 species), followed by outdoor-owned cats (20 species) and indoor cats (15 species). When accounting for undetected species using the Chao1 estimator, the asymptotic species richness was highest in feral cats (56.67, 95% CI: 38.00 – 84.48), followed by outdoor-owned cats (32.12, 95% CI: 20.00 – 57.54) and indoor cats (21.15, 95% CI: 15.00 – 37.04) (Fig 3). Thus, while pathogen diversity was still higher on average for outdoor-owned cats than for indoor cats, it was lower than for feral cats.

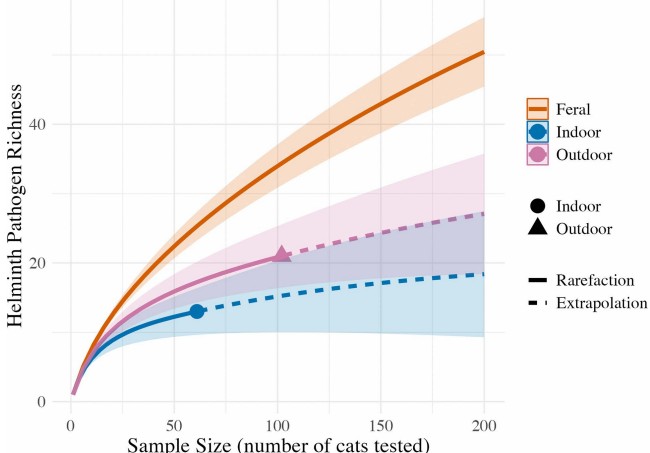

**Fig 3. Rarefaction (solid) and extrapolation curves (dotted) comparing helminth pathogen richness among indoor (blue), outdoor-owned (purple) and feral cats (orange).** Shaded areas represent 95% confidence intervals, with extrapolated values extending beyond the observed sample size to predict potential richness.

## Discussion

This study demonstrates a strong association between unsupervised outdoor roaming and the risk of zoonotic pathogen infection in domestic cats. Contrary to expectations, outdoor-owned cats exhibited infection odds that were comparable to those of feral cats, despite presumed advantages such as veterinary care, commercial diets, and other human-subsidized resources. Across all pooled pathogens, outdoor-owned cats were three times more likely to be infected than indoor-only cats. These findings are consistent with previous smaller-scale studies of owned cats [20] and further underscore the association between free roaming and elevated zoonotic infection risk. Ownership was not associated with reduced infection risk when cats were allowed to roam freely, challenging the assumption that ownership alone mitigates zoonotic risk associated with free-ranging behaviour. This assumption may contribute to the under-recognition of roaming-related consequences in public health recommendations and epidemiological investigations.

The similar all-pathogen prevalence observed in outdoor-owned and feral cats suggests that free-roaming confers a high infection risk that may accumulate rapidly, such that reduced outdoor exposure under ownership provides a limited protective effect. Owned cats that are allowed outdoors may hunt and consume infected prey, engage in interactions with wildlife or other cats, and become infested with ectoparasites, such as fleas or ticks, that serve as important vectors for zoonotic pathogens. Consistent with their substantial zoonotic burden, a large proportion of studies focused on pathogens with well-established relevance to human health, including *T. cati*, *T. gondii*, Ancylostoma spp., Cryptosporidium spp., Giardia spp., Leptospira spp., and Bartonella spp., enabling estimation of pathogen-specific odds ratios. For the majority of these pathogens, transmission is likely to be accelerated by outdoor access. *T. cati* and *T. gondii* are transmitted through both environmental and trophic routes [21,22], while Leptospira infections are typically acquired through contact with the urine or blood of infected prey, with rodents serving as major reservoirs [23].

Infection rates in prey vary according to location and prey and pathogen taxa. Rodent prey infection rates have been reported as high as 32% for *T. cati* [24], 30% for *T. gondii* [25], and 66% for Leptospira [26]. Therefore, even though hunting intensity and prey specialization vary among individual outdoor-owned cats [27,28], owner-perceived rates of ~1–15 wildlife kills per month would be sufficient for outdoor-owned cats to become infected through hunting, particularly given that owner reports underestimate true predation by approximately 80% [29]. These exposure levels are sufficient to rapidly

exceed infection thresholds, helping to explain the comparable odds of infection observed between outdoor-owned and feral cats.

Pathogen-specific patterns were consistent with known transmission routes. Bartonella is efficiently transmitted via fleas and scratches, such that even limited outdoor exposure may be sufficient for infection, potentially explaining similar risk in outdoor-owned and feral cats [30]. For pathogens maintained by peridomestic reservoirs or vectors, such as Leptospira, the risk of infection was less tightly coupled to outdoor access, as rodents can enter homes and expose even primarily indoor cats. In contrast, Giardia and Cryptosporidium showed weak and uncertain associations with outdoor access, consistent with environmentally mediated transmission from multi-host reservoirs [31–33]. Finally, the higher Ancylostoma prevalence observed in feral cats is consistent with sustained, unmanaged exposure to contaminated environments in endemic regions, where free-roaming cats and dogs maintain intense transmission cycles [34].

Although isolation of a pathogen from a cat alone does not demonstrate reservoir competence, the detection of more than 100 pathogens with zoonotic potential spanning viral, helminthic, protozoal, and bacterial taxa warrants concern. Frequent interactions with wildlife and domestic animal reservoir hosts create repeated opportunities for spillover and spillback, potentially enabling cats to assume epidemiologically meaningful roles, particularly for generalist pathogens with established or emerging zoonotic potential. For example, this dataset included HPAIV H5N1 isolation from outdoor-owned cats [34,35]. Given the high mortality and morbidity associated with HPAIV among livestock and the demonstrated potential for genetic reassortment within cats, the role of domestic cats on farms or in contact with wild or domestic birds warrants targeted investigation to inform evidence-based biosecurity recommendations [36,37].

In contrast to the prevalence of overall pathogen infection, ownership appeared to reduce the diversity of helminth pathogens recovered. Outdoor access for owned cats is generally more restricted than that of feral cats, often biased toward particular times of day and seasons [29], which may limit the range of potential prey encountered. Even when permitted outdoors, owned cats typically have shorter dispersal ranges, so they will not penetrate as deeply into natural habitats as feral cats, thereby reducing their contact with the broader diversity of wildlife reservoirs and intermediate hosts found in these environments [29]. Consequently, the cumulative exposure of owned cats to environmentally transmitted or trophically acquired pathogens is likely lower, resulting in reduced helminth richness relative to feral cats.

The relative prevalence and diversity of pathogens across cat management categories have broad significance because outdoor-owned cats not only expose their owners [38–40], but also non-cat owners through fecal contamination of public areas. For scale, outdoor-owned cats in a municipality of ~12,000 households were estimated to deposit 77 tonnes of feces annually [41]. Given that each gram of feces can contain hundreds to hundreds of thousands of long-lived ocysts, it is clear that free-roaming cats can substantially amplify environmental pathogen loads [42–45]. In this study, outdoor-owned cats had three to five times the odds of shedding a pathogen in their urine or feces compared to indoor-only cats, with risk levels comparable to those of feral cats. At the pathogen-specific level, outdoor-owned cats exhibited slightly lower average odds of shedding *T. cati* than feral cats, although this difference was not statistically significant. This pattern could plausibly reflect differences in hunting behaviour or antihelmintic use; however, deworming protocols and treatment frequency were rarely reported and could not be evaluated in the analysis. Fecal samples from cats recovered over 90 distinct pathogens, including *Echinococcus multilocularis* eggs, which is a pathogen of particular note due to the severe zoonotic consequences and immediate infectivity of this cestode species, highlighting the need for enhanced surveillance to understand how domestic cats may contribute to pathogens traditionally associated with wildlife [46]. Importantly, because reservoir impact depends on both shedding rate and host abundance, the large populations and close human proximity of domestic cats may often make their pathogen cycles more consequential to human health than those of wildlife [47,48].

The public health implications of cat-associated pathogens extend beyond environmental contamination, with human exposure occurring through bites, scratches, and the return of infected prey. For highly virulent pathogens transmitted directly, such as rabies and *Yersinia pestis*, data were primarily available as mortality cases rather than from surveillance

data; however, all cat cases were outdoor cats. Cat-associated rabies transmission is a growing public health concern globally [49]. In the US, cats are the most common rabies-positive domestic animal [50], constituting 2.5 times the exposure risk relative to wildlife in Pennsylvania [51], and in Colombia, cats are the primary source of human rabies transmissions [52]. Unfortunately, large rabies exposure datasets rarely document the lifestyles of cats that test positive, limiting their ability to inform risk-mitigation strategies. Outdoor-owned cats also, secondarily, facilitate human exposure to zoonotic viruses, with returns of prey items such as rabies-positive bats [53] and shrews and mice infected with previously undescribed and potentially zoonotic orthoreovirus and Jeilong viruses, respectively [17,18]. Domestic cats' proclivity to hunt bats and rodents is particularly problematic for spillover risk, as both taxonomic groups have been identified as high-risk reservoirs for emerging diseases [54], making complacency towards these interactions short-sighted [53,55].

There are several limitations and caveats of this study that constrain the precision with which we can assess the zoonotic risk associated with free-roaming cats. First, there is general underreporting of outdoor access of owned cats in the literature. Of the 604 papers that met the eligibility criteria, only 435 provided sufficient information on the lifestyle of owned cats to be included in the analyses. Furthermore, owner reports of outdoor access may be unreliable because owners may consider a cat that roams outside unsupervised, but sleeps in the house to be an 'indoor' cat or alternatively, a cat that has outdoor access but in a supervised manner to be an 'outdoor' cat. Although we suspect that underreporting of outdoor access is far more prevalent, both types of misreporting falsely deflate the contribution of free-roaming to pathogen infection rates in domestic cats. Further complicating the issue of lifestyle classification is that there may be country-level differences in what the public perceives as an indoor versus an outdoor cat, which could be a complex result of pet-ownership norms and no-roaming initiatives. Pathogen abundance and diversity also vary strongly with climatic, latitudinal, and anthropogenic factors, producing distinct geographic pathogen landscapes [56] and altering the absolute risk associated with free-roaming across countries and regions. To estimate relative odds, our hierarchical models account for this variation by including country, study, and pathogen as random effects; however, residual geographic structure may remain within the pooled pathogen estimates. Even for single-pathogen analyses, apparent infection risk is likely to be influenced by study-specific and geographic factors such as host abundance and density, highlighting the importance of incorporating these variables into explanatory models.

Beyond classification uncertainties and geographic variability, diagnostic sensitivity and research emphasis further influence apparent risk profiles. For pathogens that are not excreted, detection relies entirely on pathogen-specific tests guided by *a priori* expectations, such that pathogens that are not explicitly targeted will go undetected. Even for pathogens being actively shed in feces and morphologically identifiable by microscopy, such as helminths and protozoa, single-time-point sampling can result in low detectability due to intermittent shedding or low shedding intensity. False-negative rates are likely to be especially high for cryptic or novel pathogens or those shed at abundances below the detection limits of microscopy. Although microscopy was used for the majority of fecal helminth prevalence studies, molecular detection methods have much higher sensitivity. For example, comparative methodological studies have shown that *T. gondii* prevalence was estimated as zero using microscopy but reached approximately 25% when the same samples were analyzed by molecular methods [57]. However, even targeted molecular detection is limited by the scope of specific primers. Primer-independent, shotgun metagenomic sequencing avoids this limitation by not relying on pathogen-specific amplification, enabling more comprehensive characterization of pathogen communities and supporting more robust zoonotic surveillance [58].

The pathogens identified here span a wide range of public health importance, from rare but highly lethal infections, such as rabies virus, to highly prevalent pathogens, such as *T. gondii*, and many others for which human health impacts remain incompletely characterized [56]. The full population-level impact of common zoonoses is often only revealed through sustained, large-scale investigation, including the recognition of chronic or delayed sequelae, as has been the case for *T. gondii* [56]. Consistent with this pattern, the clearest lifestyle-associated risk signals in this synthesis were observed for pathogens that have received sustained research attention and for which sufficient data exist to support

robust inference. Effective zoonotic surveillance requires both broad pathogen coverage and strategic attention to domestic animals, which experience among the highest levels of routine contact with humans.

Although a multifaceted approach is needed to manage cat-associated zoonoses, limiting free-roaming remains the most impactful and pragmatic intervention to mitigate pathogen exposure. Restricting unsupervised free-roaming reduces opportunities for pathogen exposure through interactions with wildlife and free-roaming domestic animals, limits environmental pathogen loading via fecal shedding, and prevents human exposure through the return of infected prey. In contrast, preventive interventions such as vaccination or anthelmintic administration, while effective for specific pathogens, are limited in scope, do not provide coverage for many known zoonoses, do not address emerging agents, and face persistent barriers related to compliance and cost [59]. Furthermore, prophylactic anthelmintic use in free-roaming pets without addressing sources of exposure is inconsistent with anthelmintic stewardship principles, risking the development of helminth resistance [60] and contributing to ecosystem contamination with topical pesticide residues [61].

Free-roaming domestic cats represent a quintessential One Health challenge, with interconnected consequences for biodiversity, animal welfare, and human health. Unlike many One Health dilemmas that require balancing ecological or health costs against economic or social benefits, unsupervised outdoor access for owned cats offers limited demonstrable benefits while imposing substantial ecological and public health costs. Notably, unrestricted outdoor access is not considered essential for feline welfare or the human–animal bond [62]. Claims that free-ranging cats provide ecological benefits through rodent control lack empirical support, as cat-associated predation has been shown to redistribute, not suppress rodent populations or even exacerbate rodent persistence and livestock disease [63–66].

The close human association of owned outdoor cats, which contributes to their disproportionate importance in zoonotic transmission, also renders their impacts comparatively tractable through public outreach and regulatory intervention. Importantly, as free-roaming cats are increasingly recognized not solely as a conservation concern but as a public health and One Health issue, this reframing has the potential to broaden stakeholder engagement and generate greater societal support for shifts in norms surrounding pet responsibility and management. The feasibility of mitigating these risks through policy is demonstrated by a range of international legislative approaches that explicitly manage feline outdoor access to achieve public health and conservation objectives [67]. Examples include large-scale, integrated strategies such as the Cat Plan 2021–2031 in the Australian Capital Territory [68], as well as municipal bylaws such as Calgary's Responsible Pet Ownership Bylaw [69]. These models illustrate potential pathways for other jurisdictions, including the European Union, to address persistent misalignment between nature conservation frameworks and companion animal management [70,71]. While many zoonotic spillover pathways are diffuse and difficult to characterize, the contribution of free-roaming cats is comparatively direct and mechanistically well understood. Addressing this pathway through evidence-based policy, therefore, represents a tangible opportunity to reduce zoonotic risk while safeguarding biodiversity, reinforcing the application of an integrated One Health approach to companion animal management.

## Methods

### Systematic review data collection

We conducted a systematic review in accordance with PRISMA (Preferred Reporting Items for Systematic Reviews and Meta-Analyses) guidelines to compile a comprehensive global database on the prevalence of zoonotic pathogens in domestic cats. We searched Web of Science and PubMed using the terms and grammatical variants "cat," "parasite," and "zoonosis," which were selected *a priori* to align with the zoonosis-focused objective of the study while avoiding systematic bias related to cat lifestyle. Searches were restricted to studies published from 1980 to the present. Non-English studies were screened using Google Translate, and studies for which automated translation did not permit reliable data extraction were excluded (S1 Fig, S2 and S3 Tables). Additional studies were identified through reference lists of retrieved articles and several relevant reviews. Criteria for inclusion into the database were as follows: infections were natural and not experimental, both positive and negative cases were reported, and data from

PLOS Pathogens

owned and feral cats were not reported as a single group. We extracted prevalence counts, pathogen taxa, pathogen detection methods, and locations for each included study. We also extracted the author-reported lifestyle of the cats and categorized each case into five classifications: indoor-owned, outdoor-owned, unreported lifestyle-owned, shelter, and feral cats. To minimize misclassification bias [20], cats were only classified as 'indoor-owned' if the authors explicitly classified them as indoor. Any owned cat described as having unsupervised outdoor access, including 'indoor-outdoor' cats or those with 'partial' access were categorized as 'outdoor-owned.' To ensure the robustness of the comparative analysis, cases where lifestyle was ambiguous or described as 'restricted' without further detail were excluded from lifestyle-specific analyses and retained only for general prevalence descriptions. Similarly, shelter cats with unknown histories were excluded from the lifestyle and rarefaction analyses to prevent the conflation of indoor and outdoor risk profiles. Only data from indoor-owned, outdoor-owned, and feral cats were included in analyses. National-level estimates for the proportion of cats with outdoor access were compiled for 86 countries. These data were extracted from the primary study set, where available. For countries with multiple studies, an average value was calculated to represent the national trend.

## Statistical analyses

The effect of cat lifestyle on pathogen prevalence was estimated using generalized linear mixed models fitted via a Bayesian approach, implemented in the R package MCMCglmm [72]. Infection data were modelled as counts of positive and negative individuals to scale for differences in sampling effort. We first ran the analyses with all pathogens pooled, with cat lifestyle and detection method as fixed effects and study, pathogen and country of study as random effects. For pathogens: *T. gondii,* Bartonella spp., *Toxocara cati*, Leptospira spp., Ancylostoma spp., and Cryptosporidium spp., sufficient data were available to calculate pathogen-specific estimates. All models assumed a multinomial distribution and were run for 400,000 iterations, with the first 100,000 iterations discarded as burn-in to allow chain convergence, and a thinning interval of 100 iterations applied to reduce autocorrelation among retained samples. For the random effects, we used the priors as implemented in MCMCglmm (V = 1 and nu = 0.02). Beta coefficients from the model were used to calculate the odds ratios for the different management classes. To evaluate the influence of prior specification on our results, we conducted a prior sensitivity analysis by fitting the same model under weak (v = 0.02), moderately informative (v = 1), and diffuse (v = 0.5) inverse-Gamma priors for the random effects. We calculated posterior overlap scores for the key parameter cat management, which exceeded 96% across all model pairs (weak vs. moderate: 96.3%; weak vs. diffuse: 97.7%; moderate vs. diffuse: 96.6%). This high degree of consistency demonstrates that the model inference and empirical data are robust to varying prior assumptions. Model convergence was assessed using trace plots, effective sample sizes, Gelman–Rubin diagnostics (R̂), and Geweke diagnostics. For all models, fixed-effect parameters showed R̂ values close to 1.0, high effective sample sizes (>6,000), and no evidence of non-stationarity, indicating adequate convergence (S4 Table, S2a–S2h Fig).

Rarefaction and extrapolation were used to compare the diversity of helminth pathogens recovered from fecal samples of cats associated with indoor, outdoor, and feral lifestyles. Data for rarefaction only included countries with samples from all three lifestyles. These analyses were performed using the R package iNEXT, with 95% confidence intervals calculated using bootstrap resampling [73].

## Supporting information

**S1 Table. Summary of pathogen prevalence (%) with 95% confidence intervals and total number of domestic cats sampled (n), stratified by lifestyle category (feral, outdoor-owned, indoor, shelter, and unknown).** Cells are left blank where no prevalence data were identified for a given pathogen–lifestyle combination.
(DOCX)

**S2 Table. Convergence diagnostics for fixed-effect parameters from all Bayesian models.** For each parameter, Gelman–Rubin statistics are reported as the point estimate of $\hat{R}$ ($\hat{R}$) and its upper confidence limit ($\hat{R}$ upper), along with effective sample size (ESS) and Geweke diagnostics.
(DOCX)

**S3 Table. Complete reference list of all studies included in S1 Data.** All data were extracted by a single reviewer (AW) between June and December 2024. All studies listed met the predefined eligibility criteria for inclusion in the systematic review, and all extracted variables required to reproduce the analyses are provided in S1 Data.
(DOCX)

**S4 Table. Studies identified during the literature search but excluded from data extraction and analysis, with reasons for exclusion.**
(DOCX)

**S5 Table. Posterior means and odds ratios (with 95% credible intervals) for cat lifestyle contrasts in all-pathogen models restricted to studies conducted after 2000 and after 2010.**
(DOCX)

**S1 Fig. PRISMA flow diagram illustrating the study selection process for the systematic review of zoonotic pathogens in free-roaming domestic cats.**
(PDF)

**S2 Fig. Trace plots for key fixed effects from the Bayesian mixed-effects model evaluating a) all zoonotic pathogens, b) Ancylostoma, c) Bartonella, d) Cryptosporidium, e) Giardia, f) Leptospira, g) *Toxocara cati,* h) *Toxoplasma gondii* infection risk.** Plots show posterior samples for the intercept and cat lifestyle effects (outdoor-owned and feral cats), demonstrating adequate mixing and stationarity after burn-in.
(PDF)

**S1 Data. Global dataset of pathogen prevalence in domestic cats by lifestyle.**
(XLSX)

**S1 Code. Markdown file for analyses conducted.**
(RMD)

## Acknowledgments

We gratefully acknowledge the efforts of all authors whose published work allowed us to perform these global analyses.

## Author contributions

**Conceptualization:** Amy G. Wilson, Scott Wilson, Peter P Marra, David R Lapen.

**Data curation:** Amy G. Wilson, David R Lapen.

**Formal analysis:** Amy G. Wilson, Scott Wilson, David R Lapen.

**Funding acquisition:** David R Lapen.

**Investigation:** Amy G. Wilson.

**Project administration:** David R Lapen.

**Visualization:** Amy G. Wilson.

**Writing – original draft:** Amy G. Wilson, Scott Wilson, Peter P Marra, David R Lapen.

**Writing – review & editing:** Amy G. Wilson, Scott Wilson, Peter P Marra, David R Lapen.

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
