## [Decision Letter · Decision Letter 0]

20 Nov 2025

Outdoor roaming of owned cats elevates risk of zoonotic pathogen exposure: A global synthesis.

PLOS Pathogens

Dear Dr. Wilson,

Thank you for submitting your manuscript to PLOS Pathogens. After careful consideration, we feel that it has merit but does not fully meet PLOS Pathogens's publication criteria as it currently stands. Therefore, we invite you to submit a revised version of the manuscript that addresses the points raised during the review process.

We look forward to receiving your revised manuscript.

Kind regards,

Bersissa Kumsa, DVM, MSc, PhD

Academic Editor

PLOS Pathogens

Edward Mitre

Section Editor

PLOS Pathogens

Editor-in-Chief

PLOS Pathogens

orcid.org/0000-0003-2946-9497

Editor-in-Chief

PLOS Pathogens

orcid.org/0000-0002-7699-2064

**Additional Editor Comments :**

Dear Authors,

The reviewers have completed their evaluation of your manuscript. I encourage you to revise and resubmit your work, ensuring that all reviewer comments are thoroughly addressed. Please incorporate the feedback carefully and provide a detailed, point-by-point response that clearly outlines every change made in response to the reviewers’ suggestions.

In addition, kindly correct all typographical and grammatical errors, and ensure that the manuscript is prepared in full compliance with the journal’s formatting and submission guidelines.

We look forward to receiving your revised submission.

**Journal Requirements:**

At this stage, the following Authors/Authors require contributions: Amy G. Wilson, Scott Wilson, Peter P Marra, and David R Lapen. Please ensure that the full contributions of each author are acknowledged in the "Add/Edit/Remove Authors" section of our submission form.

Potential Copyright Issues:

i) Figure 1. Please confirm whether you drew the images / clip-art within the figure panels by hand. If you did not draw the images, please provide (a) a link to the source of the images or icons and their license / terms of use; or (b) written permission from the copyright holder to publish the images or icons under our CC BY 4.0 license. Alternatively, you may replace the images with open source alternatives. See these open source resources you may use to replace images / clip-art:

ii) Figure 1. Please (a) provide a direct link to the base layer of the map (i.e., the country or region border shape) and ensure this is also included in the figure legend; and (b) provide a link to the terms of use / license information for the base layer image or shapefile. We cannot publish proprietary or copyrighted maps (e.g. Google Maps, Mapquest) and the terms of use for your map base layer must be compatible with our CC BY 4.0 license.

iii) The following Figure contains a logo or branding: S1. We are not permitted to publish this under our CC-BY 4.0 license, even with permission. We ask that you please remove or replace it.

6) We note that the Data Availability Statement mentioned in the manuscript is different from that provided in the online submission form. The Data Availability statement in the online submission form is currently as follows: 'Data will be deposited into Dryad.' While the one in the manuscript states 'Data compiled for this study is available on figshare .'

We also noted that you uploaded a dataset on the online submission form as file type 'Other'.

Please provide a complete Data Availability Statement in the submission form, ensuring you include all necessary access information.

Note: The following file is currently uploaded as file type 'Other', which is not viewable by the reviewers: Wilson_et_al_PLOS_PathData.csv. Please change the file type to 'Supporting Information' and include a legend in the manuscript if you wish it to be included in review.".

7) Please provide a completed 'Competing Interests' statement, including any COIs declared by your co-authors. If you have no competing interests to declare, please state "The authors have declared that no competing interests exist". Otherwise please declare all competing interests beginning with the statement "I have read the journal's policy and the authors of this manuscript have the following competing interests:"

8) As required by our policy on Data Availability, please ensure your manuscript or supplementary information includes the following:

**Reviewers' Comments:**

Reviewer's Responses to Questions

**Part I - Summary**

Reviewer #1: This manuscript addresses a highly relevant One Health issue through an impressive synthesis of global-scale data. The authors compiled a remarkably large dataset (174,178 individuals across 88 countries), analyzed pathogen prevalence under different management regimes, and reached conclusions with direct implications for public health, wildlife ecology, and responsible pet ownership policy. The writing is clear, the importance of the issue is well motivated, and the statistical approach is generally appropriate. The study has strong potential for publication after some revisions, particularly regarding methodological transparency and the way certain assumptions and limitations are framed.

Reviewer #2: This manuscript summarizes results of a systematic meta-analysis of pathogen prevalence in domestic cats and compares disease risk in owned, roaming cats to indoor and feral cats. The research paper is well-designed, organized and covers an important topic with significance to cat, human and wildlife health. It elucidates information from the literature that should be used to educate owners and inform management of free-roaming cats.

Reviewer #3: Overall, this is an important area to research. The strengths include a well written paper with few grammar errors, and using moderately recent references.

The main concern is that this paper provides too vague of a conclusion while attempting to analyze too many infectious species and countries. The dataset does not clearly state about how old the included studies are, as this could affect the identifiable pathogens (e.g., novel or more recently identified pathogens, diagnostic method developments which would impact the ability to identify pathogens). To expand, there would be a major difference of diagnostic ability and reliability with identifying helminths between PCR and manual microscopic evaluation, even between human readers (i.e., a trained clinical pathologist versus a recent graduate who may not be as experienced).

The dataset of the identified pathogens is also lacking, particular the indoor pathogens, which would be a bias against indoor cats. Out of the all the pathogens in table S1, only 12 were analyzed of the indoor population. There is a difference between unreported and negative, so it would be unfair to use the lack of indoor pathogens as a comparison.

There are also major assumptions that weaken the study. The first is the assumption is that indoor cats are being provided with appropriate veterinary care. Secondarily, that the risk for each pathogen is equal globally, when there is a significant variation of pathogen risk between countries and even between states/provinces.

While understanding the importance of outdoor cats and zoonotic spillover is needed in the current literature, I unfortunately believe that this study is over-generalizing the population while not providing sufficient data to support the claims.

**Part II – Major Issues: Key Experiments Required for Acceptance**

Please use this section to detail the key new experiments or modifications of existing experiments that should be absolutely required to validate study conclusions.required to validate study conclusions.required to validate study conclusions.required to validate study conclusions.

Reviewer #1: NONE

Reviewer #2: N/A

Reviewer #3: 1) Compile a more complete list of indoor and owned outdoor cat pathogens.

2) Use a different and smaller subset of cases where the method of identification is uniformed/known.

3) Either include more countries or limit the review to a smaller subset of the globe for more accuracy.

**Part III – Minor Issues: Editorial and Data Presentation Modifications**

Reviewer #1: Lines 56–71: The One Health framework is introduced effectively, with appropriate references. However, the focus shifts rapidly from wildlife to domestic animals without fully articulating the conceptual bridge. To include 1–2 sentences explicitly linking domestic free-roaming species to pathogen flow networks, perhaps referencing Wells et al. 2020 or Gibb et al. 2020.

Lines 69: I think that you should add this important reference as example to support your sentence: ”ectoparasites”. I would like to suggest:

Ancillotto, L., Studer, V., Howard, T., Smith, V. S., McAlister, E., Beccaloni, J., ... & Mori, E. (2018). Environmental drivers of parasite load and species richness in introduced parakeets in an urban landscape. Parasitology Research, 117(11), 3591-3599.

Lines 72–96: The authors frame free-ranging cats/dogs as “conduits” well. However, the Introduction currently emphasizes wildlife-to-human spillover more than human-to-wildlife or wildlife-to-domestic dynamics. To clarify early that this study examines risk amplification pathways, not only wildlife-origin spillover.

Lines 97–110: The hypothesis is clearly stated. However, the authors phrase the management gradient as categorical. In reality, management (indoors → supervised outdoors → unrestricted outdoors) may be continuous. Acknowledge that lifestyle categories may vary in enforcement and duration.

Lines 113–131: The result that outdoor-owned cats have infection odds comparable to feral cats is compelling and well communicated. However: The authors should present absolute prevalence differences, not only odds ratios; The large variability among countries (OR 0.8–16.5) likely arises from lifestyle misclassification and vector presence but is not discussed here. To add a short text linking Figure 3 variation to: differences in rodent exposure; climate/latitudinal parasite gradients; cultural norms of cat care.

Lines 136–145: The rarefaction analysis is appropriate and well presented.

However, diversity is influenced not only by exposure but by diagnostic method heterogeneity. Microscopy underdetects cryptic helminths. To add a cautionary note acknowledging that observed diversity differences may be partly methodological.

Lines 146–158: The authors state that ownership does not reduce infection risk for outdoor cats.

This is justified by results, but the tone could imply causality rather than correlation. To rephrase to:

“Ownership did not correlate with reduced infection risk when cats were allowed to roam freely.”

Lines 170–206: The discussion of predator-prey pathogen transmission pathways is strong.

However, behavioral variation among cats should be acknowledged: some cats hunt often; others rarely hunt; provisioning and age strongly influence hunting behavior.

Reference: Loyd et al. 2013, Kays et al. 2020 on variability in hunting intensity.

Lines 210–244: Well written and compelling. One improvement: emphasize policy feasibility—feline outdoor access legislation exists in Germany, Australia, Finland—provide examples.

Lines 294–310: The PRISMA approach is appropriate. Search terms (“cat,” “parasite,” “zoonosis”) are too narrow and likely to underrepresent viral surveillance studies; vector-borne pathogen surveys; wildlife-predation-associated transmission research. To include additional search strings such as “Felis catus” + “pathogen”, “domestic cat” + “infection prevalence”, “helminth”, “protozoa”, “vector-borne”.

Lines 305–310: This classification (indoor-owned / outdoor-owned / unowned) is central, yet relies entirely on author-reported lifestyle, which may be ambiguous or culturally variable. Comparison with other literature such as Chalkowski et al. 2019 highlighted strong misclassification biases in global cat lifestyle reporting; Hall et al. 2020 demonstrated that even “indoor cats” may spend up to 2–6 hours outdoors depending on region. State explicitly: whether ambiguous lifestyle categories were excluded or reclassified; how cases of “partially outdoor access” were handled.

Lines 312–326: The use of Bayesian GLMMs (MCMCglmm) is appropriate for hierarchical data.

However: the prior specification (V=1, ν=0.02) may be insufficiently justified; model convergence diagnostics are not described; the decision to treat all pathogens as a pooled group assumes biological equivalence, which is unlikely. To add: trace plots, ESS, and Gelman-Rubin diagnostics to supplement S.I.; pathogen-level modeling for additional taxa where possible (e.g., Anaplasma, FeLV, FIV, Aelurostrongylus abstrusus), which exist in literature with sufficient prevalence reporting.

Lines 315: All the R codes used in this study must be added in the supplementary materials (…well commented)..

Reviewer #2: Introduction

Line 57 It may not be necessary for the audience of this journal but does One Health need a brief explanation?

Line 73 This first sentence in P2 should have a citation.

Was there any indication of how often owned cats receive veterinary care in these countries?

Results

Line 117 So only a fraction of papers reported outdoor access % for their study country?

Line 138 What is the most common type of helminth pathogen by group? Does it pose risk of zoonotic transmission?

Discussion

Line 168 This sentence should have a citation.

Line 187 Should this paragraph (p 4) be moved up before p 2?

Should the discussion address the most prevalent pathogens that are problematic (easily transmissible, symptomatic etc.) for humans?

Methods

This should report years searched, range for publications.

Figure 3 The risk of infection in just 2 of the 4 high countries is addressed in the discussion. Why might it also be high in Chile and Transylvania?

Table 1 supplement

There are no shelter or unknown columns yet that is mentioned in table description.

This may also need more description or explanation as it appears the prevalence for some pathogens is 100% of the tested sample? Maybe sample size needs to be included?

Reviewer #3: 1) Expand on line #74, on how owned animals leads to regular human contact and potentially reduced vigilance toward zoonotic disease risk. I thought that the hypothesis was that owned animals receive a higher degree of vigilance and veterinary care?

2) Expand on line #221, what is the importance of echinococcus multilocularis eggs that was necessary to specifically mention?

3) Expand on limitations of language barrier to the papers included in the study.

4) Additional references for line #234, stating that "cats acting as primary animal source of human exposure in some locations" but only adding in a reference for a very specific region of the world. It would be helpful that if you make this claim, to add literature to support it more globally.

5) Define burn-in and thinning in the statistical analyses.

6) Methods to go before results? The flow appears off with the methods being after the discussion.

7) Clarify on line #121 onwards, is the odds of infection for ALL pathogens or just bartonella, leptospirosis, toxoplasma, and toxocara?

8) In figure 3, a location is labeled as "Europe", but there are individual countries that are referred to in Europe e.g., Finland, Greece, Germany, Ireland, Netherlands.

PLOS authors have the option to publish the peer review history of their article (what does this mean?). If published, this will include your full peer review and any attached files.). If published, this will include your full peer review and any attached files.). If published, this will include your full peer review and any attached files.). If published, this will include your full peer review and any attached files.

...

Reviewer #1: No

Reviewer #2: No

Reviewer #3: No

**Figure resubmission:**

**Reproducibility:**



---

## [Editor Report · Decision Letter 1]

22 Jan 2026

Outdoor roaming of owned cats elevates risk of zoonotic pathogen exposure: A global synthesis.

PLOS Pathogens

Dear Dr. Wilson,

Thank you for submitting your manuscript to PLOS Pathogens. After careful consideration, we feel that it has merit but does not fully meet PLOS Pathogens's publication criteria as it currently stands. Therefore, we invite you to submit a revised version of the manuscript that addresses the points raised during the review process.

We look forward to receiving your revised manuscript.

Kind regards,

Bersissa Kumsa, DVM, MSc, PhD

Academic Editor

PLOS Pathogens

Edward Mitre

Section Editor

PLOS Pathogens

Editor-in-Chief

PLOS Pathogens

orcid.org/0000-0003-2946-9497

Editor-in-Chief

PLOS Pathogens

orcid.org/0000-0002-7699-2064

**Additional Editor Comments:**

Dear Authors,

The reviewers have completed their evaluation of your manuscript. I encourage you to revise and resubmit your work, ensuring that all reviewer comments are thoroughly addressed. Please incorporate the feedback carefully and provide a detailed, point-by-point response that clearly outlines every change made in response to the reviewers’ suggestions.

In addition, kindly correct all typographical and grammatical errors, and ensure that the manuscript is prepared in full compliance with the journal’s formatting and submission guidelines.

We look forward to receiving your revised submission.

**Journal Requirements:**

1) Some material included in your submission may be copyrighted. According to PLOSu2019s copyright policy, authors who use figures or other material (e.g., graphics, clipart, maps) from another author or copyright holder must demonstrate or obtain permission to publish this material under the Creative Commons Attribution 4.0 International (CC BY 4.0) License used by PLOS journals. Please closely review the details of PLOSu2019s copyright requirements here: PLOS Licenses and Copyright. If you need to request permissions from a copyright holder, you may use PLOS's Copyright Content Permission form.

Potential Copyright Issues:

i) Figure 1. Please confirm whether you drew the images / clip-art within the figure panels by hand. If you did not draw the images, please provide (a) a link to the source of the images or icons and their license / terms of use; or (b) written permission from the copyright holder to publish the images or icons under our CC BY 4.0 license. Alternatively, you may replace the images with open source alternatives. See these open source resources you may use to replace images / clip-art:

ii) Figure 1. Please (a) provide a direct link to the base layer of the map (i.e., the country or region border shape) and ensure this is also included in the figure legend; and (b) provide a link to the terms of use / license information for the base layer image or shapefile. We cannot publish proprietary or copyrighted maps (e.g. Google Maps, Mapquest) and the terms of use for your map base layer must be compatible with our CC BY 4.0 license.

**Reviewers' Comments:**

**Figure resubmission:**

**Reproducibility:**



---

## [Decision Letter · Decision Letter 2]

8 Apr 2026

Dear Dr Wilson,

We are pleased to inform you that your manuscript 'Outdoor roaming of owned cats elevates risk of zoonotic pathogen exposure: A global synthesis.' has been provisionally accepted for publication in PLOS Pathogens.

Best regards,

Edward Mitre

Section Editor

PLOS Pathogens

Edward Mitre

Section Editor

PLOS Pathogens

Sumita Bhaduri-McIntosh

Editor-in-Chief

PLOS Pathogens

orcid.org/0000-0003-2946-9497

Michael Malim

Editor-in-Chief

PLOS Pathogens

orcid.org/0000-0002-7699-2064

Reviewer Comments (if any, and for reference):

Reviewer's Responses to Questions

**Part I - Summary**

Reviewer #2: This will make an important contribution to the literature on free-roaming cats. The manuscript is well-written and author revisions addressed all of my previous questions and concerns. The additional citations are also helpful.

**Part II – Major Issues: Key Experiments Required for Acceptance**

Please use this section to detail the key new experiments or modifications of existing experiments that should be absolutely required to validate study conclusions.required to validate study conclusions.required to validate study conclusions.required to validate study conclusions.

Reviewer #2: Research methods are more clearly explained and rigorous. I do not have suggestions for further revision.

**Part III – Minor Issues: Editorial and Data Presentation Modifications**

Reviewer #2: Figures and supplemental data look great.

PLOS authors have the option to publish the peer review history of their article (what does this mean?). If published, this will include your full peer review and any attached files.). If published, this will include your full peer review and any attached files.). If published, this will include your full peer review and any attached files.). If published, this will include your full peer review and any attached files.

...

Reviewer #2: No

---

## [Editor Report · Acceptance letter]

Dear Dr Wilson,

We are delighted to inform you that your manuscript, "Outdoor roaming of owned cats elevates risk of zoonotic pathogen exposure: A global synthesis.," has been formally accepted for publication in PLOS Pathogens.

Best regards,

Sumita Bhaduri-McIntosh

Editor-in-Chief

PLOS Pathogens

orcid.org/0000-0003-2946-9497

Michael Malim

Editor-in-Chief

PLOS Pathogens

orcid.org/0000-0002-7699-2064